



# Spatially variable hydrologic impact and biomass production tradeoffs associated with Eucalyptus cultivation for biofuel production in Entre Rios, Argentina

Azad Heidari[1], David Watkins, Jr.[1], Alex Mayer[1], Tamara Propato[2], Santiago Verón[2], Diego de Abelleyra[3]

[1]Department of Civil and Environmental Engineering, Michigan Technological University, Houghton, 49931, USA
[2]INTA, Instituto Nacional de Tecnología Agropecuaria, Argentina
CONICET, Consejo Nacional de Investigaciones Científicas y Técnicas, Argentina
FAUBA, Facultad de Agronomía de la Universidad de Buenos Aires, Buenos Aires, Argentina
[3]INTA, Instituto Nacional de Tecnología Agropecuaria, Argentina, Buenos Aires, Argentina

*Correspondence to*: David Watkins, Jr. (dwatkins@mtu.edu)

**Abstract.** Climate change and energy security promotes using renewable sources of energy such as biofuels. High woody biomass production achieved from short rotation intensive plantations is an appealing strategy that is growing in many parts of the world. However, broad expansion of bioenergy feedstock production may have significant environmental consequences. This study investigates the watershed-scale hydrological impacts of eucalyptus plantations for energy production in a humid subtropical watershed in Entre Rios province, Argentina. A Soil and Water Assessment Tool (SWAT) model was calibrated and validated for streamflow, leaf area index (LAI), and biomass production cycles. The model was used to simulate various eucalyptus plantation scenarios that followed physically-based rules for land use conversion (in various sizes and locations in the watershed) to study hydrological effects, biomass production and the green water footprint of energy production. SWAT simulations indicated that the most limiting factor for plant growth was shallow soils causing seasonal water stress. This resulted in a wide range of biomass productivity throughout the watershed. An optimization algorithm was developed to find the best location for eucalyptus development regarding highest productivity with least water impact. Eucalyptus plantations had higher evapotranspiration rates among terrestrial land cover classes; therefore, intensive land use conversion to eucalyptus caused a decline in streamflow, with February, January and March being the most affected months. October was the least-affected month hydrologically, since high rainfall rates overcame the canopy interception and higher ET rates of eucalyptus in this month. Results indicate that, on average, producing 1 kg of biomass in this region uses 0.8 $m^3$ of water, and the green water footprint of producing 1$m^3$ fuel is approximately 2150 $m^3$ water, or 57 $m^3$ water per GJ of energy, which is lower than reported values for wood-based ethanol, sugar cane ethanol and soybean biodiesel.



## 1 Introduction

Using sources of renewable energy, such as biofuels, may result in cleaner, cost-competitive alternatives to fossil fuels (Winjobi et al., 2018; Sekoai et al., 2019). Cellulosic crops, crop residues and woody biomass are promising bioenergy sources because they have shown to produce similar fuel yields per feedstock mass as first-generation biofuels such as corn-based

ethanol (Lynd et al., 1991; Tilman et al., 2009). Short-rotation harvest of woody biomass is considered a major advance in bioenergy because of high rates of biomass production (Guerra et al., 2014). Eucalyptus is the most widely planted hardwood genus in the world, covering more than 19 million hectares (Binkley and Stape, 2004). Eucalyptus is highly productive (for example, >35 m$^3$ biomass/ha/year found by Albaugh et al., 2013), has a short rotation length of six to eight years, and has use as lumber and pulp (Dougherty and Wright, 2012). Many parts of the world are experiencing a rapid increase in eucalyptus

plantations for biofuel (Gonzalez et al., 2011a). There are several bioenergy products from eucalyptus, including cellulosic biodiesel and ethanol (Gonzalez et al., 2011b) and wood pellets for direct heating or electricity generation (Pirraglia et al., 2010).

According to Stape et al. (2004a), eucalyptus plantations have high water use efficiency (WUE). Furthermore, fast-growing

eucalyptus are more efficient water users compared to slower growing trees (Otto et al., 2014). However, eucalyptus plantations have been reported to have high water use compared to other species (Albaugh et al. 2013; Scott 2005) and compared to the native plants that they replace (Farley et al., 2005; Ferraz et al., 2013). In fact, eucalyptus has one of the highest ET rates among trees (Farley et al. 2005; Dye 2013; Hubbard et al. 2010), due to morphological and physiological characteristics including high stomatal conductance, evergreen leaves, drought tolerance, and deep rooting (Whitehead et al., 2004). Farley

et al. (2005) observed a higher water use rate for eucalyptus by converting grassland to eucalyptus and pine plantations on a catchment scale. They concluded that converting to eucalyptus would decrease the streamflow 25%, compared to conversion to pine. Maier et al. (2017) studied short-rotation eucalyptus planting in South Carolina, USA at the plot scale and concluded that eucalyptus had a 40% higher transpiration rate compared to pine. However, little is known about eucalyptus cultivation impacts on specific hydrologic components, i.e. baseflow vs. surface runoff, and seasonality.

Proper site selection for biofuel-related land use conversion can be crucial for sustainably managing resources in a watershed (Cibin and Chaubey, 2015). An appropriately selected biofuel crop planted at a suitable location can reduce water quality impacts of biofuel development projects (Robertson et al., 2008; Parish et al., 2012). Spatial allocation of biofuel crops has been studied on different scales, from a national level in China using geographic information systems (Zhang et al., 2017) and

at a watershed scale using optimization methods (Parish et al., 2012; Cibin and Chaubey, 2015; Herman et al., 2016; Femeena et al., 2018). However, the spatial variations in biomass production across the watershed are typically neglected. Biomass yield can vary significantly in cases where soil depth, soil quality, precipitation or temperature change across the watershed.





Sustainable biofuel production with minimal hydrologic and water pollution consequences can be achieved through scientific assessments of regional feedstock development impacts at the watershed scale (Gopalakrishnan et al., 2009; Engel et al., 2010; Watkins et al., 2015; Heidari et al., 2019a). Developing proper management practices to achieve high water use efficiency, while minimizing negative environmental impacts, requires quantification of eucalyptus water demand. To fully understand
impacts of eucalyptus development on water resources, their growth cycle and water use should be studied in more detail at sub-watershed scales.

Hydrological models have been used globally to study hydrological impacts of biofuels, especially for first-generation bioenergy crops (e.g., Schilling et al., 2008; Love and Nejadhashemi, 2011; Lin et al., 2015), but less so for second-generation
bioenergy crops (Hillard, 2017; Guo et al., 2018; Heidari et al. 2019a). SWAT is a commonly used ecohydrological, physically-based, spatially semi-distributed simulation model (Arnold et al., 2000). SWAT provides the opportunity for detailed simulations at scales ranging from tens of hectares up to watershed or river basin scales, including both hydrologic and plant growth sub-models. SWAT has been used to simulate biofuel development around the world for different crops (Babel et al., 2011; Cibin et al., 2016; Sinnathamby et al., 2017; Heidari et al., 2019a).

The goal of this study is to determine how eucalyptus-based biofuel feedstock cultivation will impact hydrological systems. Specific objectives are to assess the impacts of spatially varying patterns of eucalyptus plantation, biomass productivity, and water use for biomass production on baseflow, surface flow, and evapotranspiration. The interannual variability of hydrologic impacts is also to be evaluated, along with the explicit tradeoff between biomass production and water use. These objectives
are accomplished by calibrating and validating a SWAT model, using both hydrologic and plant growth data, for eucalyptus plantations in a watershed located in Entre Rios, Argentina. Argentina is one of the largest biofuel producing countries in the world (Statista, 2017), and the Mesopotamia region of Argentina is an appealing candidate for continuing development with eucalyptus plantations. Planting of E. grandis, which is considered to be one the most important eucalyptus species globally (Dougherty and Wright, 2012), is expanding rapidly in the region (Phifer et al., 2017).

While SWAT has been used to study hydrologic processes in various watersheds in Argentina (Troin et al., 2012; Schwank et al., 2014; Cisneros et al., 2011; Havrylenko et al., 2016; Romagnoli et al., 2017), to the authors' knowledge, this is the first application of SWAT that focuses on improving eucalyptus growth parametrization and investigating the hydrologic impacts of eucalyptus plantations for biofuel development. Considering the rapid expansion of the eucalyptus in this region of
Argentina, there is a need for more study of the water use, management and productivity of the plantations. In this work, the SWAT hydrologic and biomass growth models are calibrated and used to assess the impacts of spatially varying patterns of eucalyptus plantation, biomass productivity and water use for biomass production. The SWAT model is used to determine the feedstock stage water demand for biomass, fuel and energy production, as well as impacts on specific hydrologic components (baseflow, surface flow, and evapotranspiration) and the interannual variability of those impacts.



## 2 Methods

### 2.1 Model Setup, Calibration and Validation

The selected watershed (see Figure 1) is representative of the Argentinian Mesopotamia region. Land cover in the region typically consists of rangelands, crops such as soybeans (Modernel et al. 2016), natural forests (Espinal), orange orchards, and rivers and wetlands draining into the Uruguay River to the east. The Yuqueri Grande-Concordia hydrologic station (Base de Datos Hidrológica Integrada, 2015) was selected as the watershed outlet, and daily flow data for the period 1991-2013 was used for calibration and simulations. The contributing area to the gage was found to be 625 km$^2$, using the 30-meter resolution digital elevation model from USGS (USGS Earth Explorer, 2015) and ArcGIS 10.3. A customized streamline shapefile from the Argentina National Institute of Geography (Instituto Geografico Nacional, 2015) was used to improve the streamline delineation process.

Land use-land cover (LULC) maps from 2002-2003, 2005-2006, and 2013-2014 and soil maps were obtained from INTA. Land use land cover classifications were made with high resolution images including LANDSAT 5 and 8 with a spatial resolution of 30m from USGS (USGS Landsat Missions, 2015). For each growing season, a majority voting approach was applied considering five supervised classification methods: Maximum Likelihood, Support Vector Machines, Random Forest, LOGIT regression and Neural Networks (Waldner et al, 2016). Classes included orange orchards, agriculture, forests, eucalyptus and rangelands. Ground truth data for training and validation was compiled from different sources, including georeferenced photos, visual identification (in situ observation), georeferenced voice recordings, land owner's information, and visual interpretation of Very High Resolution (VHR) images. The overall accuracy for each LULC maps were 0.89 for 2002-2003, 0.91 in 2005-2006, and 0.95 in 2013-2014. The series of LULC maps indicated a significant decline in natural forest land (-60%) and orchards (-76%), while the area planted with eucalyptus expanded by slightly more than 100% over the 12-year period (see Table S1 for a summary of land cover change analysis). Preliminary assumptions for determining areas for biofuel development were that plantations would not compete with food crops (Paine et al., 1996) and no wetland areas would be converted. However, the land cover analysis indicated a large decrease in orange orchards and a slight variation in agriculture and rangelands over the time period evaluated.

Maximum and minimum daily temperature and daily precipitation, relative humidity and wind speed data were compiled from INTA for the Aero Concordia weather station (see Figure 1). In addition, Climate Forecast System Reanalysis (CFSR) global weather data from four more nearby stations was added to the SWAT weather database. Spatial interpolation of the climate data indicates that the area receives an average of 1220 mm of precipitation annually, with the majority of rainfall occurring during October, November and April (see Figure 2). Precipitation is higher in the eastern portion of the watershed (see Figure 1). The average annual temperature in the watershed is 19.4 ºC with slight variation across the watershed. Figure 2 shows the intra-annual variation of streamflow, temperature, precipitation, and SWAT-simulated estimates of evapotranspiration (ET)





using the Penman-Monteith method. The highest monthly average streamflow is in October, as a result of heavy rain events and average ET. Runoff efficiency, the ratio of annual stream flow to annual precipitation in the watershed, was 0.22 over the study period. Even though the months of January and February receive around 100 mm precipitation on average, they are among the lowest streamflow months due to higher temperature and ET.

ArcSWAT version 2012.10_4.19 (Winchell et al., 2013) was used for setting up the model. The watershed was divided into 8 sub-watersheds in order to assess the potential spatial variability of hydrologic impacts associated with eucalyptus cultivation. SWAT further divides the sub-basins into non-contiguous hydrologic response units (HRUs), which represent homogeneous areas within each sub-basin with unique combinations of land use, soil type and slope class. During the HRU definition,

thresholds of 0, 5 and 15% were selected for LULC, soils, and slope classes, respectively, resulting in 185 HRUs. Rivers and wetlands comprise 18% of the watershed land cover, and thus wetlands were considered in the model. Wetland functionality is described in detail in SWAT theoretical documentation (Neitsch et al., 2011) and Heidari et al. (2019).

Calibration and validation focused on both the hydrological and plant growth components of the model. The analysis was

performed for 21 years (from 1993 to 2013) to include a combination of dry and wet years in both the calibration period (1993-2005) and validation period (2005-2013). Periods with missing or unreliable data, attributed to a bridge construction project that impacted the stream gage measurements in some periods, were omitted from the goodness-of-fit calculations. The parameters controlling LAI were adjusted during the hydrologic calibration to optimize ET simulation. The calibration process included calculating the heat units for the eucalyptus in the region and changing the shape coefficients of the LAI curve.

Parameters related to LAI development stages along with potential heat units were adjusted to reflect the evergreen nature of the tree, similar to Alemayehu et al. (2017). The maximum LAI was adjusted based on field measurements (Licata, personal communication, 30 Nov 2017). The hydrologic calibration method was similar to Heidari et al. (2019a), which included separating baseflow and surface flow (Arnold and Allen, 1999). This analysis resulted in the ratio of baseflow to total flow ranging from 0.35 to 0.51 on an annual basis. The next step was to conduct a sensitivity analysis using the p-value and t-

statistic sensitivity tests in SWATCUP SUFI2 (Abbaspour, 2013). Finally, the sensitive parameters were adjusted in groups. Final adjusted parameter values are presented in Table S2 in Supplementary Materials. Performance metrics included the Nash-Sutcliffe Efficiency (Nash and Sutcliffe, 1970), coefficient of determination ($R^2$), and percent bias (Pbias) (Gupta et al., 1999). The ratio of baseflow to total flow was also required to be within the historical range.

Biomass growth is dependent on LAI and solar radiation and does not directly influence the hydrologic cycle. Therefore, it was calibrated after the LAI and hydrologic calibration. Using the final LAI parameters from the hydrologic calibration, the biomass growth was further calibrated by adjusting the radiation use efficiency and light extinction coefficient. Reported values for these parameters from De Costa and Jayaweera (1996) and Stape et al. (2004b) informed the biomass growth calibration process. It is assumed that the eucalyptus trees are planted as saplings, and the full growth cycle was simulated. In the





simulations, LAI increases year by year until it reaches the specified maximum LAI, and biomass also increases each year until the trees are harvested (Figure S1). The biomass growth calibration accounted for losses during the dormancy period, and simulated biomass at the time of harvest matched reported biomass yield in the area (INTA, 2016). Table S3 in the Supplementary Material lists the adjusted plant parameters. Full descriptions of each parameter are presented in the SWAT

theoretical documentation (Neitsch et al., 2011).

## 2.2 Modeling Scenarios

Biofuel development scenarios were formulated considering a number of variables, including the land cover types being replaced, locations of feedstock cultivation (e.g., in headwaters or downstream sub-basins), spatially variable soil fertility, and whether or not irrigation is applied. SWAT model simulations were performed for the period 1991 to 2013, using the

corresponding hydroclimatic time series. This period included a two-year warm-up period (1991-1993) to establish initial conditions, followed by a 21-year period (1993-2013) for scenario evaluation. This period allows for a range of climate conditions to be represented, as well as several harvesting rotations--specifically, the eucalyptus trees were planted at the end of August and were harvested at the end of May, with two 7-year rotations and a 6-year rotation represented in the 21-year simulation (i.e., initial planting is towards the end of the first year of the SWAT simulations). Simulated plantations were

fertilized (100 kg N/ha/year) to prevent nutrient stress. The scenarios for various land areas converted to eucalyptus consider watershed, sub-basin and HRU scales, as described in Table 1.

A bi-criteria optimization model was developed to determine Pareto-optimal combinations of sub-basins, i.e.,

$$\max_{s \in S} B \text{ and } \max_{s \in S} Q \tag{1}$$

where B is cumulative biomass production over the simulation period, Q is total streamflow, s is the sub-basin index, and S is the total set of sub-basins. The optimization procedure was based on results from the one sub-basin-at-a-time scenarios, formulated as a knapsack problem to maximize total biomass production subject to a single constraint on the allowable change in total streamflow. The tradeoff curve was generated by starting with a low level of allowable change in streamflow and then incrementally relaxing the constraint to allow more conversion.

## 3 Results and Discussion

### 3.1 Model evaluation

Comparison of simulated and observed monthly discharges, shown in Figure 3, demonstrates good performance of the hydrologic model. The Nash-Sutcliffe efficiency (NSE) of 0.55, R2 of 0.55, and Pbias of -2.9% for the entire simulation period indicate satisfactory hydrologic model performance (Moriasi et al., 2007). Table S4 shows the goodness-of-fit statistics

for the calibration and verification periods. Calibration of the biomass growth model in SWAT resulted in the most productive





HRUs matching the highest reported yields for the area, approximately 100 tons/ha/rotation. The average simulated biomass yield was 75 tons/ha/rotation, also matching the average reported values for the region (INTA, 2016) (see Figure S1 for detailed annual biomass production and LAI development simulated with SWAT). The simulated N uptake rate of 65 kg/ha/year is within the medium-high range reported by Stape et al. (2004b).

## 3.2 Analysis of watershed-scale impacts

Simulation results from all scenarios are summarized in Table 2. The intensive scenario had an average yield of 77.1 ton/ha/rotation (cumulative biomass = $9 \times 10^6$ ton). Under the intensive production scenario, streamflow was reduced at the watershed outlet on average by 28%. The surface flow was reduced by an average of 24%, with the greatest relative change in December through March (34% average decline). The average overall reduction in baseflow was 31%, with the months of January to April being the most impacted months, with an average baseflow decline of 39%. Figure 4 shows changes in monthly average total, baseflow and surface flow for the intensive scenario and the base case.

The eucalyptus plantations had the highest annual average ET rate (842 mm/year) among the terrestrial LULC classes in the basin, which was 24% higher than the average of 638 mm/year for all terrestrial LULC classes (the average over the watershed, including water bodies, was 812 mm/year). This eucalyptus ET rate is similar to what Stape et al. (2004b) reported for high-class productivity eucalyptus in Brazil for a similar climate. In the intensive scenario, the conversion increased the average annual ET over the newly converted land (319 km2) by 32% (204 mm), corresponding to a 14% increase over the watershed (625 km$^2$). The large increase in ET in the converted area was due to large areas of rangelands being replaced. Conversely, converting orange orchards did not result in a large ET difference per unit area, as orange trees have similarly high ET rates and canopy interception. Figure 5 shows the monthly average ET for the intensive scenario versus the base case during the simulation period. The substantially higher ET rate in January to April and September to December correlated to the greatest reductions in monthly streamflow shown in Figure 4.

Interannual variation in climate produced some severe decreases in streamflow due to conversion to eucalyptus. During the driest years (1999, 1995 and 2008), the average precipitation was 855 mm precipitation (compared to the mean annual precipitation of 1223 mm), and there was a 53% decline in the annual streamflow under the intensive production scenario. In wet years (25th percentile high annual precipitation), streamflow decreased only by 20%, on average. An annual precipitation of about 1200 mm was usually sufficient to saturate soils and fill the wetlands to capacity. The exceptions were 1997 (1182 mm) and 2009 (1332 mm), which had an average 38% decline in annual streamflow. These years both followed dry years, which caused large declines in soil moisture and wetland volume. Figure 6 shows the cumulative distribution of streamflow for the base case and intensive scenario over the simulation period. A significant shift downward in stream flows is observed as a result of replacing existing land cover with eucalyptus plantations, especially for the lower flows. The shift was smaller for higher flows as they are associated with heavier rainfall.





The extreme scenario had an average biomass yield similar to the intensive scenario, with 77.2 ton/ha/rotation, but produced a higher cumulative biomass ($12\times10^6$ ton) as a result of converting 83% of the watershed to eucalyptus. This conversion increased the average annual ET by 18%, causing a 37% decline in the average annual streamflow. When 434 mm/year of irrigation of eucalyptus was added to the intensive scenario, the number of water stress days decreased by 85% and the cumulative biomass production of the watershed increased to $12.3\times10^6$ ton, an increase of 36% over the non-irrigated intensive scenario.

**3.3 Variability in biomass productivity due to spatial variations in soil properties and climate**

The simulation results indicate a wide range in biomass production at the HRU scale (average area = 3 km$^2$), from 37 to 97 ton/ha/rotation for the intensive scenario. Figure 7 shows the variation in soil depth, precipitation, and yield across the sub-basins. The most critical spatially variable parameters for determining biomass productivity were precipitation and soil depth. The lowest productivities were associated with shallow soils (<500 mm deep soils), which reduce growth because the reservoir of available soil water is small, leading to water stress during low-rainfall or high-ET periods. Comparing HRUs with similar soil depths across sub-basins, different yields were mainly due to precipitation differences in these sub-basins. In wetter sub-basins, the relatively high precipitation maintained the water content of the soil, leading to a reduction in water stress. The results in Figure 7 allow comparison between the lowest biomass yield (sub-basin B) and highest biomass yield (sub-basin H) sub-basins. Sub-basin H is typical of the downstream portions of the watershed, comprising the highest soil depths and precipitation. Sub-basin B is typical of the upstream portions, where soil depths are shallower and annual precipitation is about 200 mm lower than the downstream sub-basins. In the intensive irrigated case, the additional water increased biomass yield by 50% in the upstream sub-basins.

Using results from the intensive scenario, the HRUs were sorted from the highest biomass productivity to the lowest, and high-yield HRUs were defined as those having a productivity of more than 75 ton/ha/rotation (in the upper half of the reported range of 50-100 ton/ha/rotation). The high-yield HRUs were then grouped so as to cover approximately one-third, two-thirds, and the total area of high-yield HRUs (a total area of 213 km2). The simulation results in Table 2 show that converting two-thirds of the highest yield HRUs (HY2) resulted in the highest productivity (83.7 ton/ha/rotation) among all the non-irrigated scenarios simulated in this study. Table 2 also shows, however, the high water cost per biomass for the HY scenarios. This is due to most of the high-yield HRUs being located in basins H and G, which have deep soils and high precipitation, leading to diminishing returns with respect to water use efficiency.





### 3.4 Watershed-wide trade-offs between biomass production and water consumption as a result of targeted eucalyptus cultivation

Figure 8 summarizes the tradeoffs between biomass production and streamflow impacts at the main outlet for the scenarios involving conversion of each sub-basin, one at a time. In the base case, the cumulative biomass yield was $1.6\times10^6$ ton, or an average yield of 75 ton/ha/rotation. Sub-basins F, G, and H were inferior to the other sub-basins because they had higher precipitation rates and a greater impact on streamflow at the outlet compared to the other sub-basins. In contrast, planting in sub-basins A and B produced a considerable amount of biomass with a relatively small decrease in the streamflow. This was surprising as these sub-basins had high local impacts at a sub-basin level (Table 2). However, sub-basins A and B had small watershed-wide impacts because they received lower precipitation amounts and their contribution to the total streamflow at the outlet was relatively small (see Table 2). Further investigation of sub-basins A and B helped to understand how the hydrological impacts were dependent on which land cover was replaced, as well as the presence of water bodies. Sub-basin A experienced a high local impact (at the sub-basin level) since it was dominated by rangelands and it had a small area covered by rivers and wetlands. In sub-basin B, even though the total amount of converted area was greater, local hydrologic impacts were mitigated due to a larger area in this sub-basin being covered by water (Figure 1).

Figure 9 shows the biomass production-water impact tradeoff analysis generated by the optimization model (see detailed results in Supplemental Material, Table S5). At low levels of allowable change in streamflow (less than 3%), only one sub-basin was converted at a time (i.e., E, F, C, A). As the streamflow constraint was relaxed, the model continued with the best combination of two or more sub-basins until all sub-basins were converted. During the optimization, sub-basins A and C were picked the most frequently (22 and 19 times, respectively, out of 29 solutions), even though sub-basins H and G had the highest productivity (selected 5 and 6 times each). This result is explained by the fact that the high biomass yield in those two basins came with the cost of high water consumption. In other words, the biomass production per unit of water consumption was highest for sub-basins A and C. A notable water-efficient solution, corresponding to ABC, produced $3.0\times10^6$ additional tons of biomass with only a 9.9% decrease in total streamflow relative to the base case. For comparison, the intensive scenario produced an additional $7.0\times10^6$ tons of biomass but resulted in a 28.8% decrease in total streamflow. Point ABCDE was also a critical point, as the slope of the trade-off curve steepened beyond this point due to the optimization model being forced to select sub-basins F, G and H in the rest of the solutions. Sub-basins G and H were the selected the least because of their low productivity per unit of water consumption.

### 3.5 Green water footprint

Water footprints represent the total water consumption associated with a production system, with green water defined as precipitation that is stored in the soil and available for evapotranspiration, and blue water defined as water extracted from rivers, lakes and aquifers (Hoekstra and Chapagain, 2006; Chiu and Wu 2013). Table 3 summarizes the range of water requirements for different biofuel production scenarios estimated in this study and several others. The calculations for Table 3



were based on an assumption of using the total aboveground biomass (stems, branches and leaves) with no losses (Guerra et al., 2016). Furthermore, this study only reports the water use at the farm gate, considering that total water use in the life cycle of biofuels is dominated by the feedstock production stage (Gerbens-Leenes et al., 2009). Water use values in Table 3 mostly account for rainfall and soil moisture and can be considered green water, except for a few cases that include irrigation, which

is categorized as blue water. On average, simulations conducted herein indicate a water requirement of 0.79 m$^3$ to produce 1 kg of dry biomass, or 1.26 kg of dry biomass would be produced from 1 m$^3$ of water. This value is similar to that found by Maier et al. (2017), 0.69 m$^3$ water/kg dry biomass for short rotation eucalyptus in South Carolina, USA. Moreover, Stape et al. (2004b) reported a similar but slightly lower range (0.31-0.62 m$^3$/kg wet biomass) for eucalyptus production in Brazil.

Assuming each kilogram of biomass can produce 0.32 kg of fuel (GREET, 2016), and neglecting the water used at the refinery, an average of 2150 m$^3$ of water would be used to produce 1 m$^3$ of biodiesel. Further assuming this liquid fuel would have an energy content that is similar to conventional diesel fuels, 43 MJ/kg (BP, n.d.), results in a water footprint of 57.1 m$^3$ water/GJ, or 206,000 l/MWh. A similar result from Maier et al. (2017) shows 16.1 m$^3$water/GJ for biodiesel derived from eucalyptus feedstock. The "additional water" shown in Table 3 is defined as the difference in water use (ET) between the intensive case

and the base case. This represents the direct hydrologic impact of converting land to eucalyptus, which is just 0.17 m$^3$/kg of biomass, or 14.6 m$^3$water/GJ of energy. Notably, these water use estimates for eucalyptus are orders of magnitude lower than what Dominguez-Faus et al. (2009) reported for irrigated corn and soybean. Moreover, the 57.1 m$^3$ water/GJ found in this study is significantly lower than the reported values for wood-based ethanol (Schyns et al., 2017) and ethanol from sugarcane and soybean (Rodriguez et al. 2018). This indicates that planting eucalyptus in the case study basin can be a water-efficient

method for biofuel feedstock production, especially if plantations are located on deep fertile soils which, considering the region's high average annual precipitation, will eliminate the need for irrigation.

### 3.6 Discussion of Limitations

Leaf area index is a key parameter for plant growth models. It is related to photosynthesis, water and nutrient use, rate of growth, and accumulation of dry matter (Smethurst et al., 2003; Ishak and Awal, 2007). Similarly for SWAT, LAI is a key

parameter for simulating actual ET and biomass production (Neitsch et al., 2011). LAI measurements (Maire et al., 2011; Smethurst et al., 2003) indicate the non-linear nature of LAI development over time in eucalyptus trees. However, in SWAT, LAI increases at a constant rate until it reaches a specified maximum allowable LAI, and the annual growth rate is limited by a single parameter (number of years to maturity). This simplified model of LAI development can lead to inaccurate estimates of water use and annual incremental biomass production. LAI development in SWAT can be calibrated by changing the number

of years to maturity, but this parameter also impacts the ratio of aboveground biomass to total biomass. Thus, there may be a tradeoff between accurate modeling of aboveground biomass and total biomass production.



Another limitation in the growth model is the dormancy period. In SWAT, trees, perennials, and cool-season annuals go dormant as the day length nears the minimum for the year. Furthermore, the LAI starts to decrease to a specified minimum leaf area index during the dormant period. Both of these model assumptions are inaccurate for an evergreen tree such as eucalyptus. Despite the improvements to modeling LAI in this study (see LAI-related parameters in Table S3), a dormant period was simulated for two weeks in winter (mid-July), when the minimum day length occurs. However, the biomass growth calibration accounted for losses during the dormancy period, and simulated biomass at the time of harvest to matched reported values for the region.

**4 Conclusions**

The main objectives of this work were to study the hydrological impacts and water-biomass tradeoffs associated with the development of eucalyptus plantations for bioenergy production. A SWAT model was set up to study the hydrological impacts of biofuel development projects in a humid subtropical region of Argentina (northeastern Mesopotamia region, Entre Rios province). The model was calibrated and validated using long-term weather data and streamflow measurements, and plant growth parameters were adjusted to improve the LAI growth and biomass production predictions. The model was used to simulate 21 years of eucalyptus growth in two 7-year rotations and one 6-year rotation. A range of eucalyptus plantation scenarios were defined to evaluate hydrological effects, biomass production potential, and the green water footprint of energy production under different assumptions about land use conversion.

Hydrologic model results indicated that the ET rates of eucalyptus were the highest among the local terrestrial LULC classes in the watershed, which resulted in a decline in the streamflow the amount of which depended on the area and location of the plantations. For an intensive scenario of converting rangelands, orange orchards, and forest (62.5 % of the watershed), an average annual decline of 28% in the total streamflow (including both surface and baseflow) was simulated. The greatest decline occurred during months of February, January and March, with an average decrease of 37%.

Planting eucalyptus in different parts of the watershed resulted in a wide range of biomass productivity (37-100 tons/ha/rotation), due to variability in soil depth and precipitation across the watershed. The lowest biomass production occurred on shallow soils, where limited soil moisture storage led to more frequent water stress. The water stress was more severe if the shallow soil was located in a sub-basin with lower average annual precipitation.

Water-biomass tradeoffs resulted from the more productive plantations having higher ET rates and consequently greater impacts on water yield at the watershed outlet. To some extent, the tradeoffs could be mitigated by accounting for the land cover being replaced and the amount of water bodies in the area. The ET rate was higher for open water than all terrestrial LULC classes, making it a controlling hydrologic process for the sub-basins. Further, conversion of orange orchards had less hydrologic impact per unit area than converting rangeland, although the total area of orange orchards was small.

Based on model results, the average green water footprint of biodiesel produced from eucalyptus was 2150 m$^3$ per m$^3$ fuel, or 57.1 m$^3$ per GJ energy. These amounts are lower than or similar to reported values from other studies and crops in different parts of the world.

**Data availability**

The input data and results of the SWAT model are available at
http://www.hydroshare.org/resource/ff7fbbcb8a0a451da606f855ec391639 (Heidari et al., 2019b).

**Author contributions**

AH was responsible for data curation, formal analysis, investigation, and visualization. DW and AM supervised AH and were
responsible for conceptualization, funding acquisition, and administration of the project. TP assisted with data curation and formal analysis and provided resources. SV and DA supervised TP and assisted with data curation and resources. AH wrote the original draft, and DW, AM, TP, SV, and DA reviewed and edited the manuscript.

**Competing interests**

The authors declare that they have no conflict of interest.

**Acknowledgments**

This material is based upon work supported by the National Science Foundation - Partnerships for International Research and Education (PIRE) Program (OISE-PIRE #1243444) and Research Coordination Network (RCN) Program (CBET #1140152), as well as by the Inter-American Institute (IAI) for Global Change Research (CRN3105). We thank J. Licata for consulting on field LAI measurements and geospatial data for the study site.

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



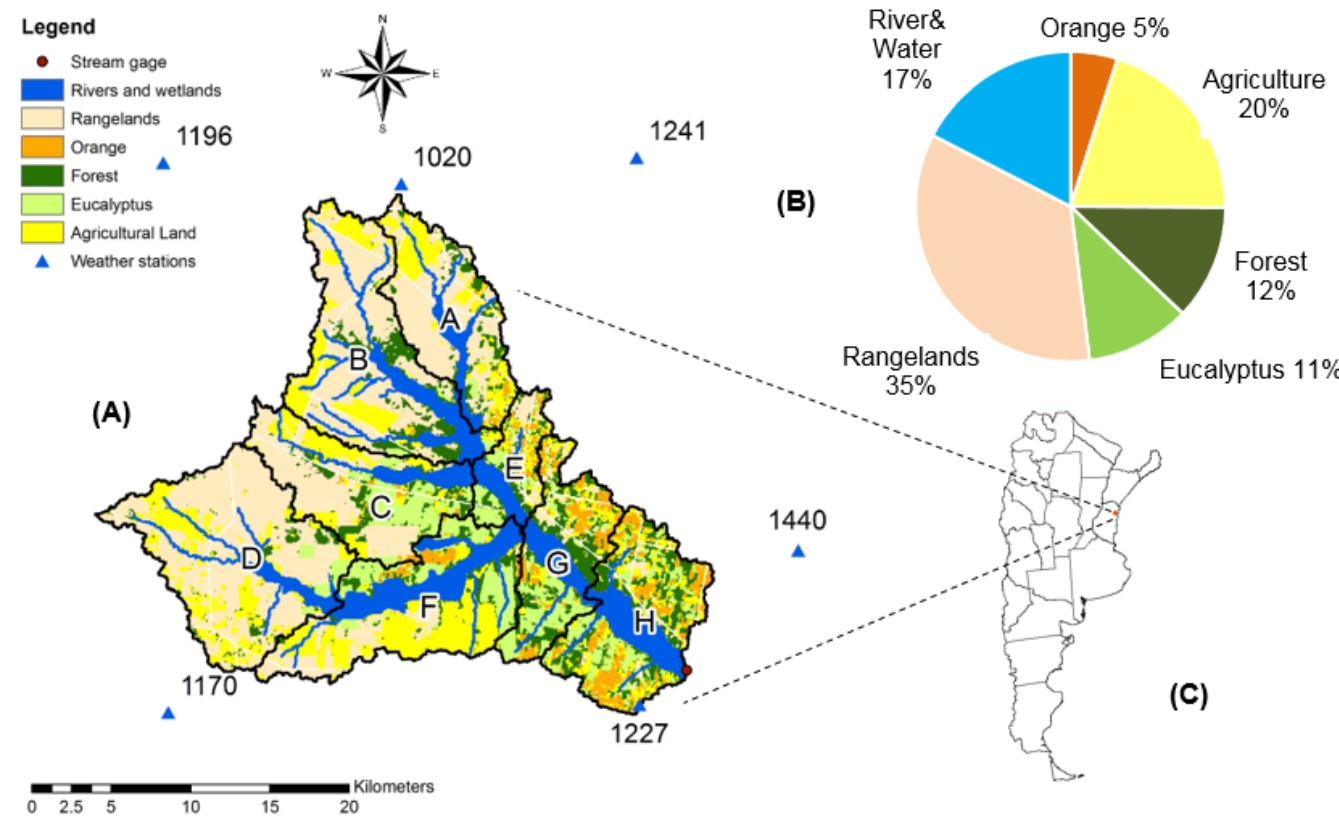

**Figure 1: A) Land use/land cover map of the watershed, locations of the sub-basins, and precipitation gauges with average annual precipitation (in mm); B) Land cover distribution in the watershed; and C) Study site location within the state of Entre Rios, Argentina.**





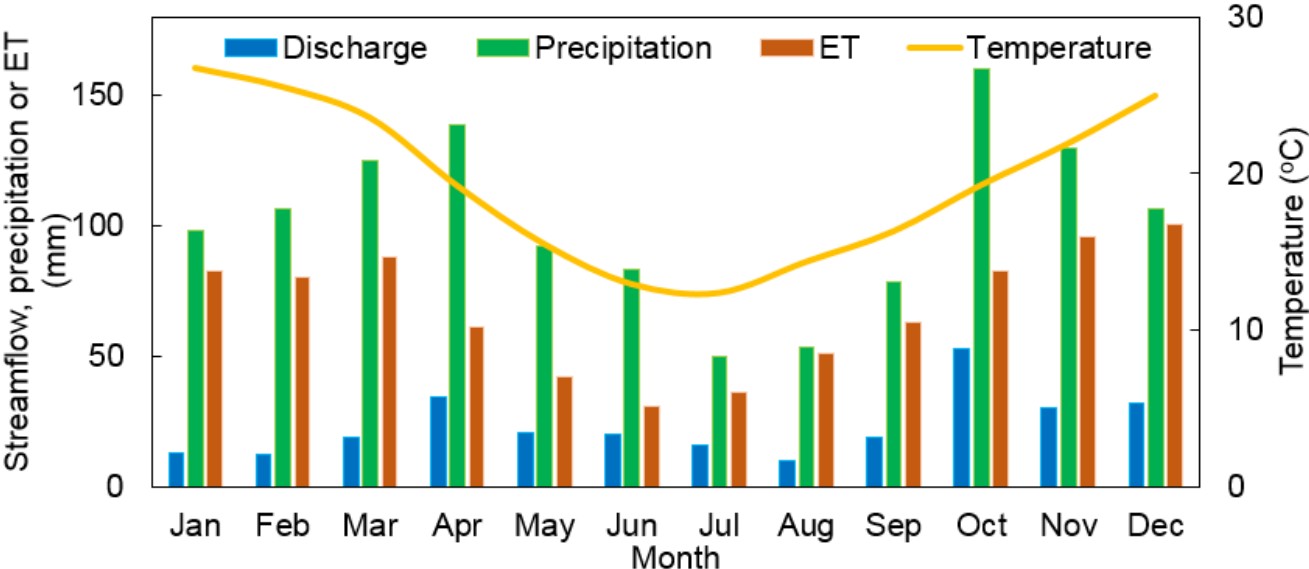

**Figure 2: Intra-annual patterns of monthly average precipitation, streamflow, temperature and actual ET (simulated).**

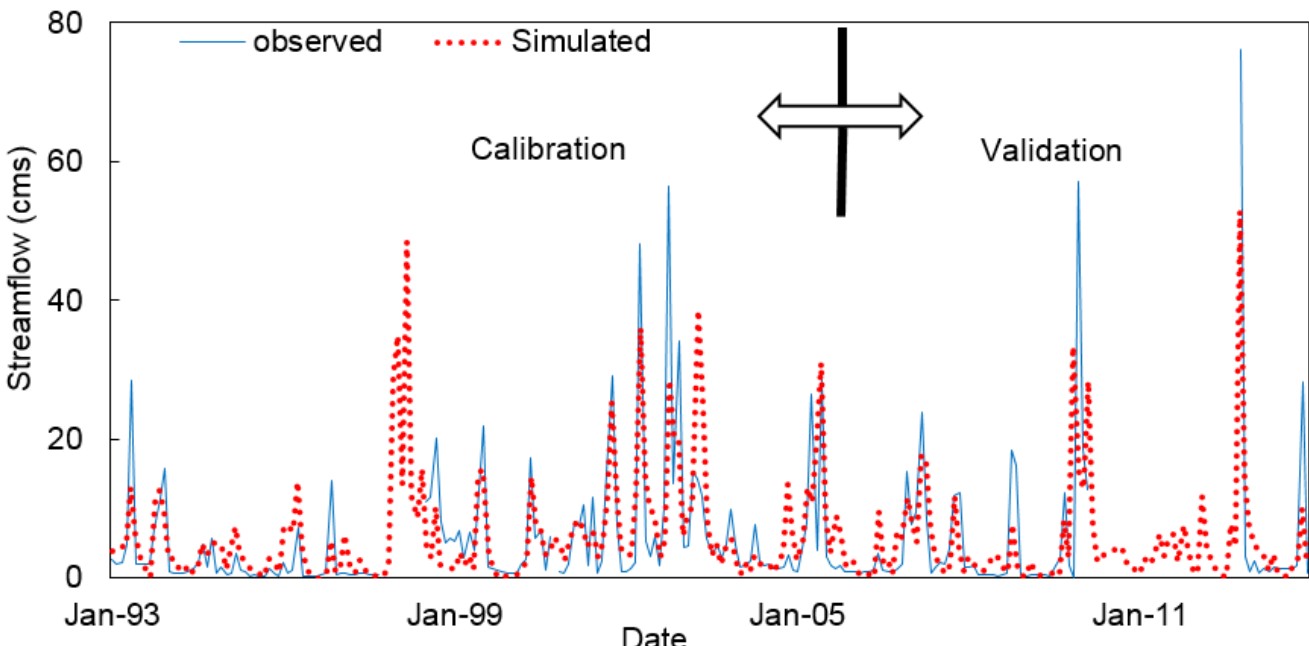

5    **Figure 3: Observed and simulated monthly streamflow during the calibration (1993-2005) and validation (2006-2013) periods.**





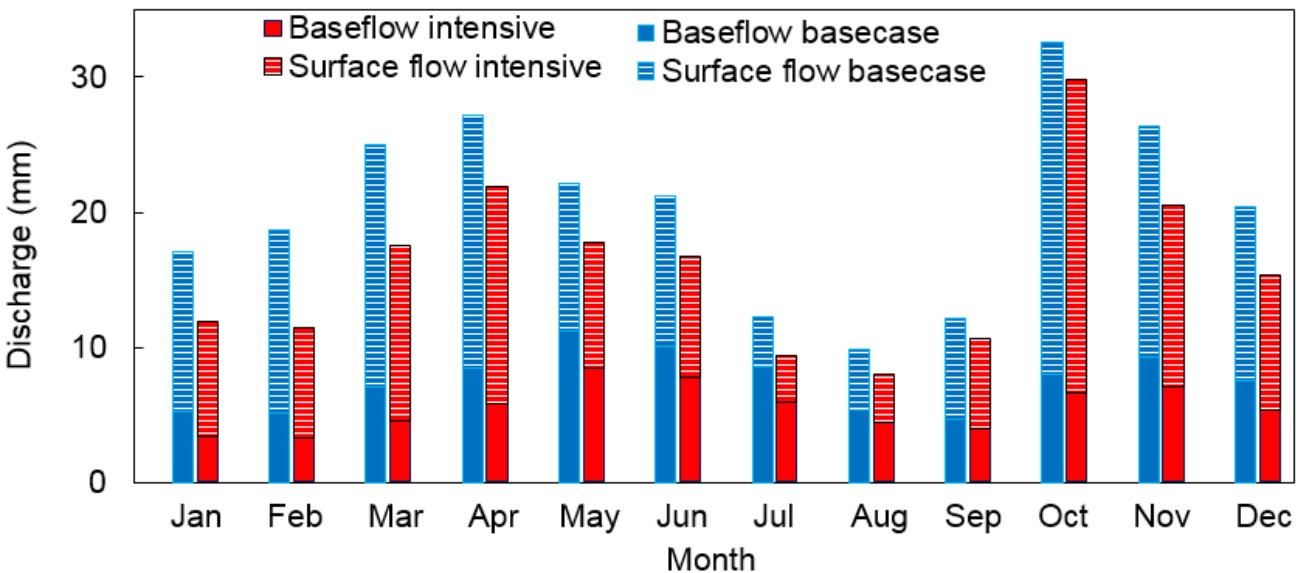

**Figure 4: Average monthly baseflow, surface flow and total flow for Base case and Intensive scenario.**

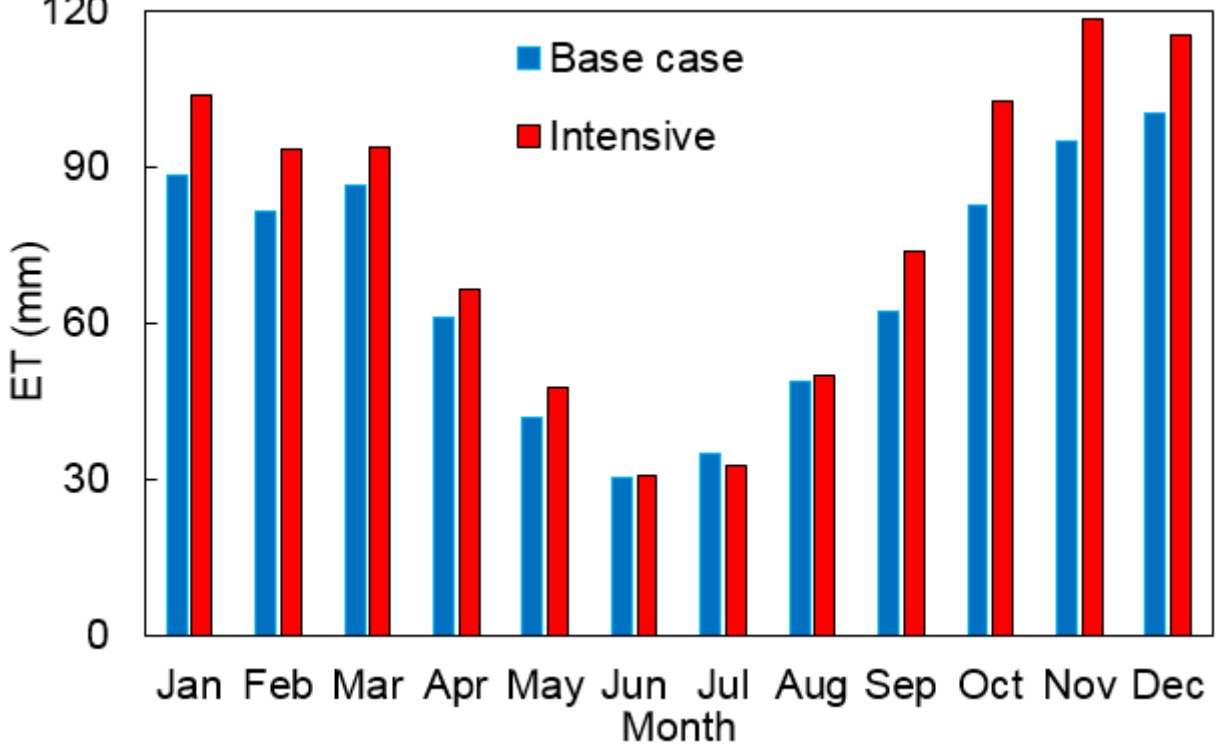

**Figure 5: Monthly average simulated ET rates for base case and intensive scenario.**

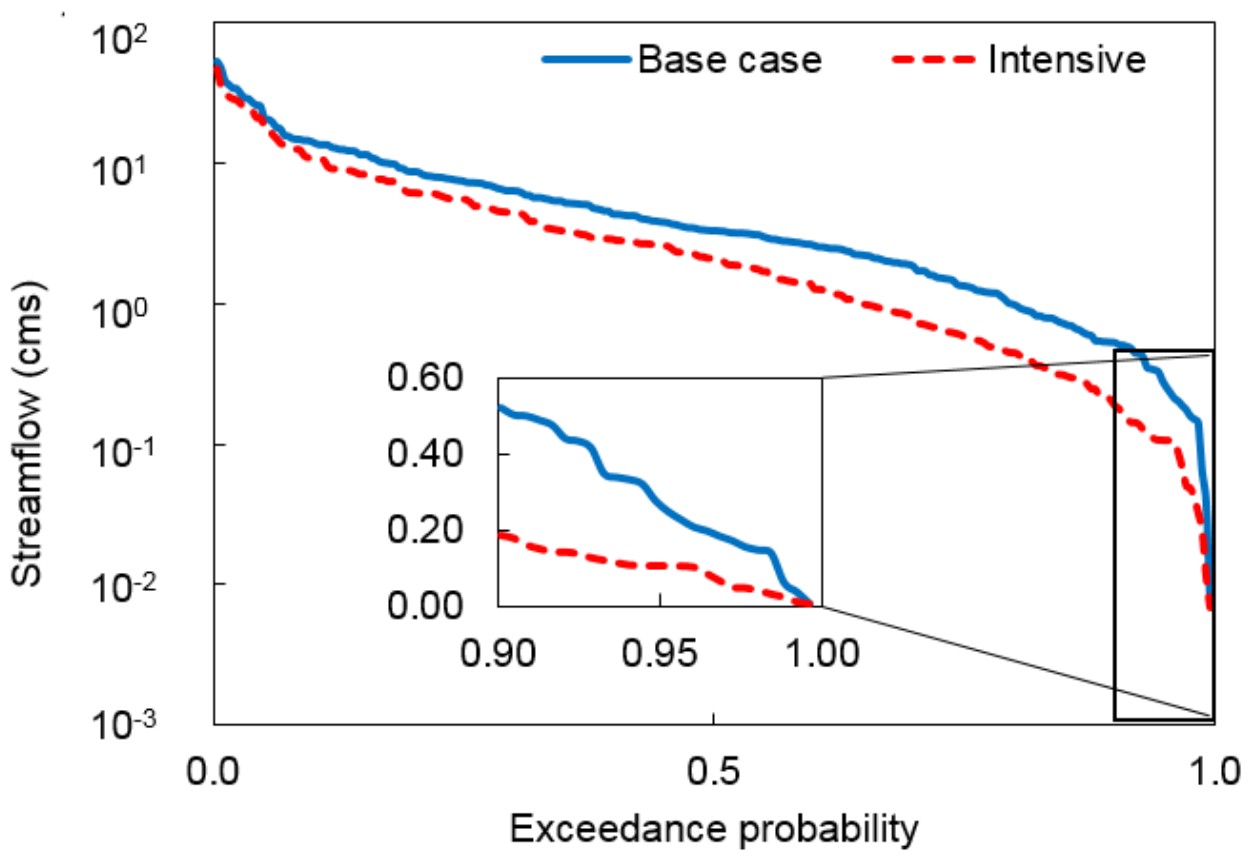

5   **Figure 6: Cumulative distribution functions of monthly streamflow for the full simulation period (1993-2013) under Base case and Intensive scenarios. Inset shows the low-flow tails.**



5    **Figure 7: Map of soil depth, precipitation and yield variation in the watershed.**



5 **Figure 8: Cumulative biomass and relative streamflow changes for the Base case and one sub-basin at a time scenarios. Bubbles are scaled to the area of eucalyptus plantations in each scenario.**





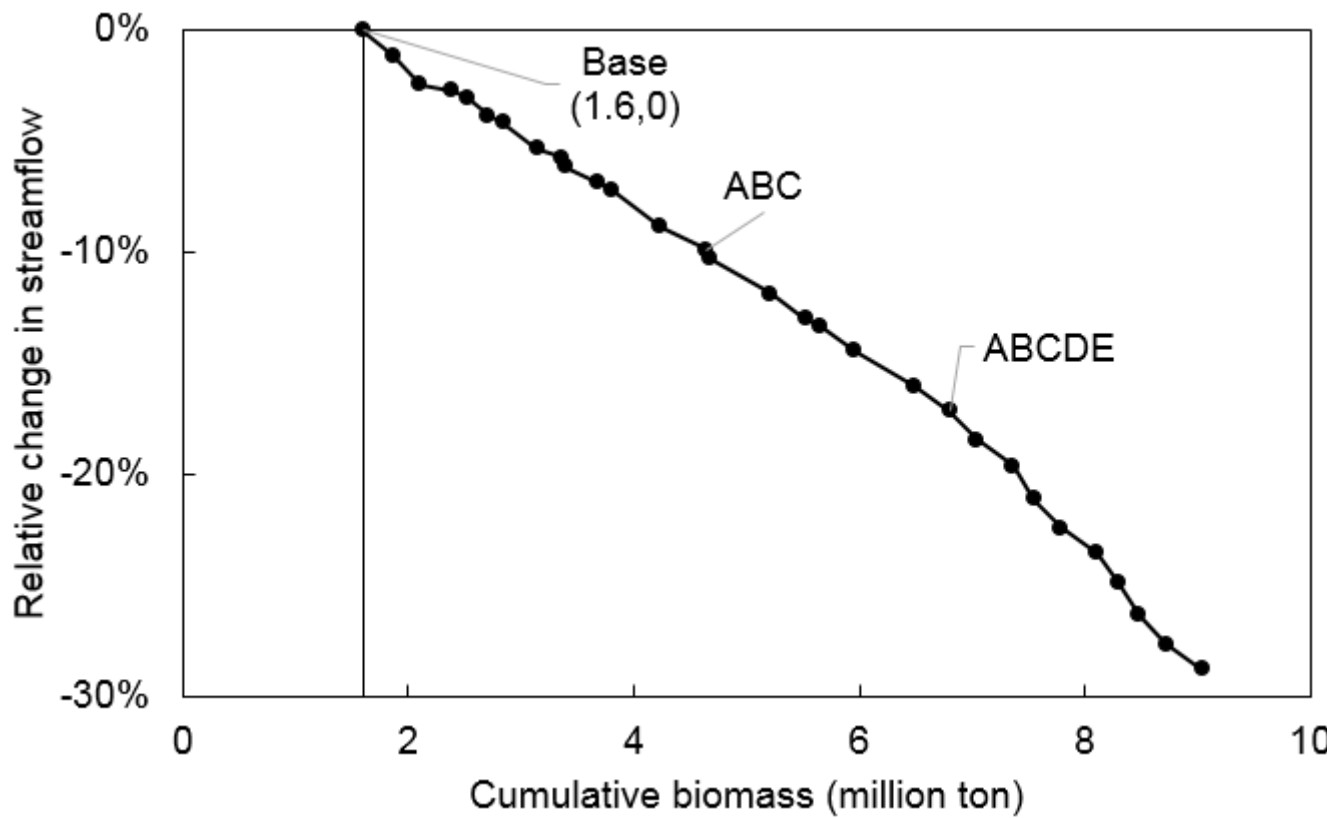

**Figure 9: Tradeoff between biomass production and hydrologic impacts at the watershed scale. The Base case and combinations of sub-basins with relatively high productivity per unit of water consumption are highlighted.**



**Table 1: List of eucalyptus development scenarios simulated in SWAT.**

| Treatment (code) | Scale of conversion | Description | Area converted (km$^2$) |
|---|---|---|---|
| Base case | Watershed | LULC is based on 2002 conditions, with 69 km$^2$ (11% of total area of 625 km$^2$) already planted with eucalyptus. | -- |
| Intensive (Int) | Watershed | All LULC classes except crops and wetlands are converted to eucalyptus. | 391 |
| Extreme (EX) | Watershed | All LULC classes except wetlands are converted. | 517 |
| Intensive irrigated (IntIr) | Watershed | All LULC classes except crops and wetlands are converted to plantations with irrigation. | 391 |
| A, B, C, D, E, F, G, H, AB, AC, etc. | Sub-basin | All LULC classes, except crops and wetlands, are converted in one sub-basin at a time and in combinations of sub-basins. | Varies (28-119) |
| HY1 | HRU | The top one-third of high-yield HRUs, defined as those with productivity >79 ton/ha/rotation, are converted. | 126 |
| HY2 | HRU | The top two-thirds of high-yield HRUs are converted. | 172 |
| HY3 | HRU | All high-yield HRUs are converted. | 219 |





**Table 2: Summary of biomass production and hydrologic impacts for each scenario.  All changes are relative to the Base case.**

| Treatment | Code | Fraction watershed with eucalyptus | Cumulative biomass produced (10⁶ ton) | Yield of biomass (ton/ha /rotation) | Annual ET watershed (mm) (% change) | Annual streamflow at outlet (mm) (% change) | Additional water[a] per additional biomass (mm/10⁶ ton) | Contribution to flow at outlet in Base case | Change in water yield at sub-basin |
|---|---|---|---|---|---|---|---|---|---|
| Base case | | 11.0% | 1.6 | 75.1 | 812 ( - ) | 248 ( - ) | - | - | - |
| Basin A | A | 17.6% | 2.5 | 76.5 | 826 (1.7%) | 240 (-3.0%) | 8.9 | 6.7% | -41.1% |
| Basin B | B | 21.2% | 2.8 | 71.0 | 832 (2.4%) | 238 (-4.2%) | 8.3 | 12.1% | -34.1% |
| Basin C | C | 17.2% | 2.4 | 74.3 | 824 (1.5%) | 241 (-2.7%) | 8.8 | 7.3% | -32.0% |
| Basin D | D | 23.9% | 3.4 | 75.8 | 840 (3.4%) | 233 (-6.1%) | 8.3 | 12.7% | -42.7% |
| Basin E | E | 12.8% | 1.9 | 77.6 | 817 (0.7%) | 245 (-1.1%) | 10.0 | 5.4% | -19.9% |
| Basin F | F | 15.1% | 2.1 | 74.6 | 823 (1.3%) | 242 (-2.5%) | 12.0 | 17.5% | -14.1% |
| Basin G | G | 15.4% | 2.3 | 79.6 | 825 (1.6%) | 238 (-3.9%) | 14.3 | 15.2% | -24.2% |
| Basin H | H | 16.5% | 2.5 | 80.4 | 828 (2.0%) | 235 (-5.2%) | 14.4 | 23.0% | -22.3% |
| Intensive | Int | 62.6% | 9.0 | 77.1 | 927 (14.2%) | 179 (-27.8%) | 9.3 | - | - |
| Extreme | Ex | 82.8% | 12.0 | 77.2 | 961 (18.3%) | 157 (-36.7%) | 8.8 | - | - |
| Intensive, Irrigated | IntIr | 62.6% | 12.3 | 104.9 | 1171 (44.2%) | 193 (-22.1%) | 5.1 | - | - |
| High yield 1 | Hy1 | 19.4% | 3.0 | 83.2 | 839 (3.3%) | 228 (-8.0%) | 14.3 | - | - |
| High yield 2 | Hy2 | 25.8% | 4.0 | 83.7 | 853 (5.0%) | 219 (-11.7%) | 12.1 | - | - |
| High yield 3 | Hy3 | 34.1% | 5.2 | 82.6 | 872 (7.3%) | 208 (-15.9%) | 11.1 | - | - |

a)  Computed based on the change in streamflow at the watershed outlet





**Table 3. Water requirement for biomass, fuel and energy production estimated from this study and others.**

| Scenario | Water per biomass (m³/ kg) | Water per fuel (m³/ m³) | Water per energy (m³/ GJ) |
|---|---|---|---|
| Average | 0.79 | 2148 | 57.1 |
| Lowest yield | 0.81 | 2207 | 58.7 |
| Highest yield | 0.75 | 2073 | 55.1 |
| Irrigated[a] | 0.85 | 2328 | 61.9 |
| Additional water[b] | 0.20 | 551 | 14.6 |
| Other studies: | | | |
| Maier et al. (2017) – Eucalyptus biodiesel | 0.69 | | 50.1[c] |
| | 0.35 (Wet) | | |
| Schyns et al. (2017) - Wood-based ethanol | | 2260 | 97.0 |
| Dominguez-Faus et al. (2009) - Corn ethanol, irrigated | | | 630-2408 |
| Dominguez-Faus et al. (2009) - Soybean ethanol, irrigated | | | 3861-77490 |
| Wu et al. (2012) Corn stover ethanol | | 760-1000 | |
| Rodriguez et al. (2018) - Sugarcane ethanol | 0.2 | | 76.0 |
| Rodriguez et al. (2018) - Soybean biodiesel | 1.5 | | 242 |
| Rodriguez et al. (2018) – Soybean, 2nd harvest biodiesel | 2.5 | | 411 |
| Chiu and Wu (2013) – Wood residue ethanol | | 212-1705 | |
| Stape et al. (2004b) – Eucalyptus biomass | 0.31-0.62 (Wet) | | |
| Babel et al. (2011) - Oil palm biodiesel | | | 110 |
| Gerbens-Leenes et al. (2009) - Sugarcane ethanol | | | 108 |
| Gerbens-Leenes et al. (2009) - Soybean biodiesel | | | 394 |

a) Includes green and blue water.

b) Computed as increased ET relative to the base case land use/land cover; refer text for details.

c) Not directly given by the author. Calculated with this study's assumptions.

