# Peer review of "Spatially variable hydrologic impact and biomass production tradeoffs associated with Eucalyptus cultivation for biofuel production in Entre Rios, Argentina"

_Hydrology and Earth System Sciences, 2019_

## Referee Comment (RC1) · Anonymous Referee #1 · 25 Jul 2019

General comments: In this study, the SWAT was calibrated/validated and applied to evaluate the impacts of spatially varying patterns of eucalyptus plantation, biomass productivity, and water use for biomass production in a watershed in Argentina. The research findings from this study provided guidance for woody biomass tree planting and water resource management. This paper is well organized. Method description, scientific results, and conclusions are presented clearly and concisely. The number and quality of references are appropriate.

Specific comments: In my opinion, calibration and application of the SWAT model it-
self in this study do not carry sufficient scientific meaning. The creativity and importance of this study are not enough for publication in this journal. Without an in-depth discussion on physical processes represented by the calibrated SWAT and how to apply the research results, this research will be difficult to be implemented by potential stakeholders/policymakers. Moreover, simple discussion on water consumption by eucalyptus plantation in section "3.5 Green water footprint" does not represent the water footprint well. The impacts of bioenergy crop growth on sediment and nutrient losses in the watershed are important but not covered in this research. Additionally, tradeoffs between the costs of eucalyptus plantation and potential environmental impacts were not considered in this research, either. More in-depth discussion should be included to support the interpretations and conclusions. What is the novel idea this manuscript provided to scientific knowledge? Please describe it and use your results and discussion to support it. Does this manuscript develop a new methodology, tool, or theory? What can readers learn from this research, and how can they adopt it in other regions?

Technical corrections: page 1: line 13: "promotes" or "promote"? line 15: add "the" before "broad expansion" line 13: add "the" before "highest" and "least" line 25: add "," before "March"

page 2: line 3: add "," before "and woody biomass" line 8: add "a" before "use" line 15: change "are" to "is" or add "trees" after "eucalyptus" line 25: change "i.e." to "i.e.," line 32: add "," before "or"

page 3: line 5: add "the" before "impacts of" line 30: add "," before "and productivity" line 32: add "," before "and water use"

page 4: line 2: add "," before "and Validation" line 13: add "," before "including" line 17: add "," before "and rangelands" line 27: add "," before "and wind speed data" line 31: add "," before "and April" page 5: line 2: "stream flow" or "streamflow"? Please be consistent through the manuscript.

page 6: line 27: add "the" before "good performance". What is the definition of "good

performance"? What are differences between good and satisfactory goodness-of-fit statistics? line 28: improve the format of "R2"

page 7: line 27: change "An annual precipitation" to "Annual precipitation" line 32: add "the" before "existing"

page 8: line 23: delete "a" before "productivity" line 25: improve the format of "km2"

page 9: line 4: add "the" before "conversion"

page 10: line 24: add "," after "Similarly"

page 11: line 11: change "limitation in the growth model" to "limitation of the growth model" line 15: change "were defined" to "was defined"

page 19: Figure 3: change "observed" to "Observed"

page 20: The font size of Figure 5 looks like larger than that of Figure 4. Please be consistent through the manuscript.
* * *

---

## Referee Comment (RC2) · Anonymous Referee #2 · 12 Dec 2019

This ipaper tackles an important issue of the influence of Eucalyptus trees, and different managements strategies, on the hydrological mass balance and flows of catchments. It is well written and a good description of the model application and evaluation is given. But, the application of Eucalyptus is a challenge in SWAT, as it might be the first implementation (as stated by the authors). This would, in my opinion, need a stronger check on the Eucalyptus growth and water use. SWAT usually struggles with tropical forests (as acknowledged in line 1 of page 10, SWAT simulates dormancy in forests), and would for that reason not be suitable for this study.

[Figure]

Some issues to be considered:

1. How are the seasonal dynamics simulated? What is the influence of the simulation of dormancy on the results? How is the interaction with the roots simulated? SWAT is typically not doing this, except if one would use REVAP parameter to mimic this process, but this is not linked to any vegetation/root parameter. How are the trees reacting to drought? 2. Are the simulation results on yields realistic (cfr line 12 of page 8)? Are the relationships with soil depth and precipitation confirmed with observations? 3. Line 17 days "The parameters controlling LAI were adjusted during the hydrologic calibration to optimize ET simulation" but I don't find any comparison or evaluation for the ET simulations. My suggestion is to provide the Hydrological Mass Balance as a check, ideally also ET is evaluated for Eucalyptus. 4. The CN values became very low, and the recharge_DP parameter is very high, and might lead to unrealistic results in the hydrological mass balances with too high deep losses (which are not going to the outlet). 5. Some details are missing. Which evapotranspiration method was used? Which routing method was used? In summary, the model needs a better check, both in the calibration of the hydrology as on the implementation of Eucalyptus in SWAT. In my opinion, SWAT in general, and the model application for this case study, is not ready to be used for scenarios on Eucapyptus plantations and this might lead to wrong conclusions.

---

## Author Comment (AC1) · 10 Jan 2020

RC1: This paper is well organized. Method description, scientific results, and conclusions are presented clearly and concisely. The number and quality of references are appropriate.

AC: Thank you for your encouragement.

RC1: In my opinion, calibration and application of the SWAT model itself in this study do not carry sufficient scientific meaning. Without an in-depth discussion on physical

processes represented by the calibrated SWAT and how to apply the research results, this research will be difficult to be implemented by potential stakeholders/policymakers.

AC: Although extensive field measurements were not available for this study, the following physical processes were checked for accuracy using literature values in this modeling work including: streamflow (separated to baseflow and surface flow), ET, N uptake, LAI development, and Biomass production. We can add some discussion of these physical processes and some additional (literature-based) justification of the calibrated parameter values (Table S1), along with some discussion of soil characteristics and geologic setting.

RC1: Moreover, simple discussion on water consumption by eucalyptus plantation in section "3.5 Green water footprint" does not represent the water footprint well.

AC: We have reported the water footprint at the farm gate level as discussed on (P10, L2). Farm-gate level water use is a well-accepted term among the biofuel research community. For example: "...this study only reports the water use at the farm gate, considering that total water use in the life cycle of biofuels is dominated by the feed-stock production stage (Gerbens-Leenes et al., 2009)." However, unlike the paper cited above, we failed to mention that we evaluated the gross production of bioenergy, rather than the net production, meaning that we did not account for energy inputs in the production chain. Neglecting energy inputs means that the water footprint will be underestimated, especially when bioenergy production systems have large energy inputs. We will add this clarification and limitation to the paper.

RC1: The impacts of bioenergy crop growth on sediment and nutrient losses in the watershed are important but not covered in this research.

AC: We agree on the importance of water quality issues. Unfortunately, water quality measurements were not available for this study, and thus sediment and nutrient losses were not reported. However, we did check that N uptake was within a reasonable range (P7, L3). If the reviewer would like to see the sediment and nutrient losses simulated

by SWAT, we can add these uncalibrated results either in the paper or supplemental information. A study of water quality impacts of eucalyptus development in this region remains as important future work, and we would be happy to add a statement to this effect.

RC1: Additionally, tradeoffs between the costs of eucalyptus plantation and potential environmental impacts were not considered in this research, either.

AC: We agree this is an important aspect in decision-making, but as with water quality, detailed economic data were not available for this study. We would be happy to mention this as important future work.

RC1: What is the novel idea this manuscript provided to scientific knowledge? Does this manuscript develop a new methodology, tool, or theory? What can readers learn from this research, and how can they adopt it in other regions?

AC: This study provides a simulation-based approach for planning and managing biomass production at the watershed scale, accounting for tradeoffs between biomass production and water use. A novel aspect is the consideration of spatial variability. As a result, producing a map such as Figure 7 provides a framework that can be applied in other contexts to locate areas with relatively high productivity and/or low hydrologic impacts. In addition, as acknowledged by the second reviewer, "to the authors' knowledge, this is the first application of SWAT that focuses on improving eucalyptus growth parameterization and investigating the hydrologic impacts of eucalyptus plantations for biofuel development." We provided a detailed calibration and parameterization for plant growth (including LAI and biomass production), and the resulting plant growth parameters may be valuable for use in other regions, with potentially small adjustments.

---

## Author Comment (AC2) · 10 Jan 2020

RC2: But, the application of Eucalyptus is a challenge in SWAT, as it might be the first implementation (as stated by the authors). SWAT usually struggles with tropical forests (as acknowledged in line 1 of page 10, SWAT simulates dormancy in forests), and would for that reason not be suitable for this study.

AC: We agree this was a challenging application of SWAT, but we attempted to make the improvements necessary to simulate the full life cycle of eucalyptus growth. For ex-

ample, the dormancy was a very brief period (only two weeks), which was accounted for during the calibration and did not have a significant effect on the ET or biomass growth. We hope to make a contribution to the literature, as further attempts for coupling forest and hydrology models are required.

RC2: It is well written and a good description of the model application and evaluation is given.

AC: Thank you.

RC2: a) How are the seasonal dynamics simulated?

AC: Seasonal discharges (both surface flows and base flows) are represented fairly well by the model, as shown in Figure 3. Data for seasonal biomass growth was not available, but growth dynamics are simulated based on solar radiation, temperature and precipitation that change seasonally.

RC2: b) What is the influence of the simulation of dormancy on the results?

AC: Although there is a short dormancy period simulated (2 weeks), we made adjustments to make sure it did not significantly affect the results. This is briefly discussed on (P6, L2): "The biomass growth calibration accounted for losses during the dormancy period, and simulated biomass at the time of harvest matched reported biomass yield in the area (INTA, 2016)." The only potential impact of the dormancy could be the slight underestimation in ET in that period.

RC2: c) How is the interaction with the roots simulated? SWAT is typically not doing this, except if one would use REVAP parameter to mimic this process, but this is not linked to any vegetation/root parameter.

AC: REVAP controls a process in which water moves into the soil zone from a shallow aquifer in response to water deficiency. This process is significant in watersheds where the saturated zone is not very far below the surface or where deep-rooted plants are growing. In our case, soils are deep (resulting in a deep saturated zone) in the eastern

parts of the watershed where eucalyptus is mostly growing. Also, relatively high average precipitation throughout the watershed prevents water deficiency. Thus, the impact of REVAP was not significant in this case study. Furthermore, changing this parameter would result in a change in ET and the overall water balance. Since the simulated ET was reasonable compared to literature values, we chose not to change REVAP from the default value in SWAT.

RC2: d) How are the trees reacting to drought?

AC: As shown in Figure S1, the trees are generally resilient to droughts that occur during this hydrologic record. We noticed some sensitivity to drought near the end of the rotation, when LAI is near a maximum (e.g., 2006), resulting in a slightly lower yield for that rotation. We can discuss this further in the revised manuscript.

RC2: 2. Are the simulation results on yields realistic (cfr line 12 of page 8)? Are the relationships with soil depth and precipitation confirmed with observations?

AC: Yes, the simulated yields are realistic, and they were calibrated based on reported values in the region. This is explained in (P7, L2). The average simulated biomass yield was 75 tons/ha/rotation, matching the average reported values for the region (INTA, 2016). The range of 70-80 tons/ha/rotation is also reasonable, and consistent with the eastern part of the watershed having more plantations, where soils are deeper and rainfall is higher.

RC2: 3. Line 17 says "The parameters controlling LAI were adjusted during the hydrologic calibration to optimize ET simulation" but I don't find any comparison or evaluation for the ET simulations. My suggestion is to provide the Hydrological Mass Balance as a check, ideally also ET is evaluated for Eucalyptus.

AC: We mention on (P7, L13): "The eucalyptus plantations had the highest annual average ET rate (842 mm/year). This eucalyptus ET rate is similar to what Stape et al. (2004b) reported for high-class productivity eucalyptus in Brazil." The value reported by

Stape et al. was 880 mm/yr, which is about a 5% difference. Note that LAI parameters were also adjusted to represent the biomass production.

RC2: 4. The CN values became very low, and the recharge_DP parameter is very high, and might lead to unrealistic results in the hydrological mass balances with too high deep losses (which are not going to the outlet).

AC: We considered these parameter values carefully in ensuring that the water balance was reasonable. As mentioned, we performed a baseflow separation and required the simulated ratio of baseflow to total flow to be within the historical range. The overall water balance of the watershed is: P = ∼1220 mm/year ET = ∼812 mm/year Stream-flow = ∼300 mm/year The Deep Recharge parameter seems high, but simulated deep recharge is around 100 mm, which is less than 9% of the precipitation. Physical justi-fication is that the eastern portion of the watershed has deep soils and drains into the Uruguay River.

RC2: 5. Some details are missing. Which evapotranspiration method was used? Which routing method was used?

AC: The ET method is Penman-Monteith (P5, L1). Penman-Monteith was selected as it is a standard method recommended by FAO for ET calculations. The routing method is the Variable Storage Routing Method, the default routing method in SWAT. We will add a sentence identifying this routing method.

RC2: In summary, the model needs a better check, both in the calibration of the hydrol-ogy as on the implementation of Eucalyptus in SWAT. In my opinion, SWAT in general, and the model application for this case study, is not ready to be used for scenarios on Eucalyptus plantations and this might lead to wrong conclusions.

AC: We agree that one would need to be extremely cautious in applying model re-sults to make decisions. This is one reason for including a section on study limitations, which will be expanded to accommodate insights from reviewers. We hope that we can

convince the reviewer that the model is adequately calibrated and validated for both hydrology and plant growth simulations. To summarize the checks that were done: 1)The hydrological model passes standard evaluation tests, including: a) Pbias of less than 10% shows the very good performance of the model on the overall water balance; b) NSE of 0.6 shows better than satisfactory (>0.5) and close to very good (>0.65) performance of the model on monthly flow (Moriasi et al., 2007). 2) ET values match reported data in the literature. 3) The ratio of baseflow to total flow (annually) is maintained within the historical range, based on a baseflow separation. 4) LAI values match the regional measured values and values reported in the literature. 5) Biomass production values match the reported values for the region (Figure S1). 6) N uptake of the trees is in the range reported in the literature. (P7, L3)